# The Psychosocial Risk Factors Evaluation and Management of Shift Personnel at Forest Harvesting

**Yana Korneeva** [1,]*  , **Natalia Simonova** [1,2]  and **Nina Shadrina** [1]

1   Department of Psychology, Northern (Arctic) Federal University Named after M.V. Lomonosov, 163002 Arkhangelsk, Russia
2   Laboratory of Labor Psychology, Faculty of Psychology, Moscow State University Named after M.V. Lomonosov, 125009 Moscow, Russia
*   Correspondence: amazonkca@mail.ru; Tel.: +7-960-004-5657

**Abstract:** The study describes psychosocial risk factors at work in relation to the general functional state of a body, working capacity and stress among shift workers at a logging enterprise in the Far North. The study involved 153 loggers with a 14-day shift period. Research methods included the psychophysiological instrumental method (variocardiointervalometry) used to assess the general functional state of a body, M. Luscher's color test to assess working capacity and stress and QNordic to assess psychosocial factors. Statistical analysis was performed using multiple stepwise regression analysis and frequency analysis. It was found that 71.1% of employees have a favorable general functional state of the body, 28.9%—unfavorable; the forest loggers who took part in the survey have a high level of working capacity and a low level of stress. It was substantiated that the general functional state of a body, stress and working capacity, measured by objective and subjective methods, are differentially interconnected with psychosocial factors at work. The general functional state of forest harvesting workers is influenced by factors of labor content, intensity and organization. The relationships with the immediate supervisor are important in order to increase the working capacity and efficiency of employees as well as their involvement in work.

**Keywords:** psychosocial factors at work; forest safety and health; forest harvesting; shift method; stress; forest mechanization

## 1. Introduction

Worldwide, forestry is among the most physically hazardous industries [1–6]. The factors of adverse environmental impact on loggers with shift labor organization include the high intensity of work, high sensory loads (for operators of logging machines), increased shift duration, the need to work in night shifts, lack of natural lighting, atmospheric pressure drops and high air humidity [7,8]. The forest harvesting process is complicated by the following features: marshy areas, lack of communications, shifting forest logging sites (they can be several kilometers from the place of residence) and the frequency of work shifts (day and night, which adds to loggers' stress. Moreover, considering the automation of labor, i.e., the transition from manual logging to the use of forest harvesting machines (harvesters and forwarders), new professional risks arise. This is confirmed by the results of studies, which found that long and/or irregular working hours have a negative impact on the psychological well-being of workers, especially due to the reduction in personal time [9–11].

The specific climatic and geographical conditions of the Far North, which increase the negative impact of production factors, and the peculiarities of the shift labor organization should be added to the above labor factors in the forest harvesting sector. Researchers have repeatedly studied the influence of climatic-geographical, industrial and social factors on the health, psychological well-being and career longevity of shift workers [12–14].

Work in such conditions is accompanied by an increased risk of developing negative functional states, health hazards and failures of occupational reliability and safety [7,8,15–17]. In this context, negative functional states, including job stress, should be considered as psychological stress or a set of negative psychophysiological responses and reactions [18] that occur when the requirements of the work environment exceed the capabilities and resources of employees, and also when the working environment does not satisfy their needs [19,20].

New Zealand scientists found that 70% of accidents in the forest industry are caused by employee errors, and 78% of cases of workers' disabilities are caused by fatigue. The authors concluded that a significant number of forestry accidents are caused by overworked workers [21]. The researchers found that 53% of surveyed forest employees worked while sick or injured, and 46% were overworked [22]. Other studies also showed that poor mental health is associated with risky and dangerous behaviors that can lead to accidents and loss of quality as a result of unfavorable events [20,23,24].

Automation of production at forest harvesting has had a positive effect on reducing the risk of injury due to a decrease in the share of physical labor and the convenience of using mechanized equipment. The accident rate for manual logging is four times higher than for mechanized forest logging, both in Louisiana [25], Sweden [26] and Russia [27]. At the same time, studies indicate the influence of the season of the year on the injuries of forest loggers: the highest intensity of accidents is observed in the summer [16] and in January, February and March [4,6,28].

Some of the literature data show that personal and organizational conditions, collectively known as psychosocial factors at work, play a large role in safety in the forestry industry [3,16,17]. The importance of identifying negative psychosocial factors and preventing harm from their impact on workers' health is emphasized in [18]. At the same time, psychosocial factors can not only harm the condition of workers and lead to errors and injuries in the workplace but also improve the well-being of staff [24–30].

The World Health Organization [31] developed a list of work-related psychosocial hazards for the European Psychosocial Risk Management System, which includes ten main factors divided into two groups: work content and work context. In order to assess these factors, questionnaires were developed and tested, including the General Questionnaire for Psychological and Social Factors at Work (QPSNordic), which was developed by the Nordic Council of Ministers, validated and standardized in four Nordic countries [32]. This tool allows for assessing the whole complex of psychosocial factors in the workplace and is actively used in various samples, including extractive industries [33–36].

Some researchers emphasize the importance of studying each block of psychosocial factors at work. It was shown that work with high demands and a low level of control is accompanied by high stress and leads to mental disorders and deterioration of physical health [37,38]. The importance of diversity in monotonous work, as well as the importance of teamwork, was studied by Boldman and Deal [39]. It was established that the feeling of social support in the workplace is positively associated with job satisfaction and organizational commitment of employees [40]. Social support, which is seen as a buffer against stressors, has a beneficial effect on health [41].

In research on the relationship between psychosocial factors at work and depression, anxiety and stress, three aspects of the model of stress at work according to the Job Demands–Control–Support (JDCS) model are widely used [42]. It was established that psychosocial factors at work are associated with emotional exhaustion in women and men. These factors include organizational justice and bullying [43]. It was proven that psychosocial factors at work have a significant impact on mental health and work capacity [44].

By summarizing various studies, Beheshtifar and Mahmoudi concluded that negative psychosocial factors contribute to the development of the following negative characteristics in workers: workplace stress and burnout, emotional exhaustion, depressive and

musculoskeletal symptoms, etc. At the same time, positive psychosocial factors contribute to increased productivity and improve the quality of working life [45].

A comprehensive study of psychosocial factors at work among builders showed that they indirectly influenced safety indicators, affecting their professional psychological state (emotional burnout and involvement). Work stress, role uncertainty, work–family conflicts and interpersonal conflicts negatively affect safety performance, contributing to burnout and impacting engagement. Autonomy and social support were positively associated with safety performance, improving work engagement and reducing burnout [46].

The literature data revealed a great danger of work in forest harvesting due to specific conditions and organization of labor, the impact of employees' functional state and well-being on labor safety (injuries, errors in work, etc.). Unfortunately, a comprehensive assessment of the diversity of psychosocial factors that affect workers at forest harvesting has not been conducted so far.

Researchers often examine the negative states and experiences of employees, which include stress, increasing the risk of injuries and accidents in the workplace of forest-harvesting workers. At the same time, it is important to assess the impact of work on the development of positive psychological states, which include working capacity. The optimal working capacity allows employees to perform their tasks efficiently and on time. In this connection, it is important to determine not only the factors influencing the development of stress and overwork to control them and reduce their impact on staff but also the factors that contribute to the improvement of working capacity and performance.

The assessment of psychosocial risk factors in conjunction with the diagnosis of the workers' state and well-being during the shift period allows us to determine the key organizational and personal conditions that affect the work safety of loggers and develop targeted recommendations for improving working conditions and increasing the efficiency of their professional activities.

The study's purpose was to identify and describe psychosocial factors at work in correlation with the general functional state of the body, stress and working capacity of shift workers at a forest harvesting enterprise in the Far North.

## 2. Materials and Methods

This research is empirical and analytical. All research methods were discussed by the ethics committee of the Higher School of Psychology, Pedagogy and Physical Culture of the Northern (Arctic) Federal University and recommended for use (protocol No. 2, 2021).

### 2.1. Procedure

**Hypothesis 1.** *The general functional state of the body, measured using the objective psychophysiological method, will have a strong correlation with the psychosocial factors of work requirements.*

**Hypothesis 2.** *Stress and working capacity, measured using the subjective psychological method, will have a close connection with psychosocial factors of social support, leadership and organizational culture.*

**Hypothesis 3.** *The frequency analysis of assessments of psychosocial factors at work will allow us to identify positive and negative factors that affect the majority of employees at a forest harvesting enterprise.*

Positive psychosocial factors, which are evaluated by the majority of employees, can be correlated with hygiene factors in Herzberger's motivational theory [47,48]. Negative psychosocial factors, which are noted by the majority of employees, require measures to reduce their impact in order to maintain workplace safety and efficiency of personnel.

The research materials were collected during a scientific expedition to a logging enterprise in the Arkhangelsk region of Russia from 18 October to 5 November 2021 and also from 18 to 21 April 2022. During the period of personnel shifts (completion of the 14-day

working period by some employees and the beginning of the 14-day working period by other employees) in their collection points (villages of Karpogory and Yasny, Arkhangelsk region), two psychologists conducted diagnostics and a projective psychological assessment of employees' functional state, as well as the survey of employees using QPSNordic. All subjects voluntarily agreed to take part in the study by providing written voluntary informed consent.

### 2.2. Sample

One hundred and fifty-three male workers aged 23 to 59 years old took part in the study (average age $39.5 \pm 0.77$ years old), with work experience from 0.5 to 34 years (average experience $6.5 \pm 0.76$ years), with fly-in-fly-out work experience from 0.5 to 23 years (average experience $6.4 \pm 0.50$ years, fly-in period duration—14 days).

The sample is representative of the general population of forest loggers in the Arkhangelsk region of the Russian Federation due to the correspondence of the demographic indicators of the sample to the demographic indicators of the enterprise as a whole and the presence of respondents of different ages, education, length of service, positions and separate structural divisions (Table 1).

**Table 1.** Demographic characteristics of the sample.

| Age | Up to 30 Years | 30 to 40 | Over 40 | |
|---|---|---|---|---|
| amount | 26 | 68 | 59 | |
| percent | 17.0% | 44.4% | 38.6% | |
| **Education** | General average | Secondary vocational | Higher | |
| amount | 57 | 85 | 11 | |
| percent | 37.3% | 55.6% | 7.2% | |
| **Work experience in position** | from 0.1 to 5 years | from 6 to 15 years | over 16 years old | |
| amount | 99 | 38 | 16 | |
| percent | 64.7 | 24.8 | 10.5 | |
| **Shift work experience** | from 0.1 to 3.4 years | 3.5 to 8.4 years | from 8.5 to 15 years | over 15.1 years |
| amount | 59 | 54 | 28 | 12 |
| percent | 38.6 | 35.3 | 18.3 | 7.8 |
| **Professional group** | logging machine operators (harvester, forwarder, motor grader) | drivers | mechanics and welders | masters |
| amount | 99 | 34 | 16 | 4 |
| percent | 64.7 | 22.2 | 10.5 | 2.6 |

The study involved loggers working on a rotational basis. The technological process at the logging site includes the following main operations: felling trees, cross-cutting, delimbing, sorting, packing, hauling, stacking and loading timber assortment onto a log truck. The labor functions performed by workers at the logging site include the management and maintenance of multi-operational forest machines (harvesters and forwarders). A harvester machine is equipped with an automated control system that performs a set of operations: grabbing, cutting, felling trees, delimbing, marking and bucking tree trunks. A forwarder machine is equipped with attachments or trailed equipment, which performs a range of operations for stacking, picking up and hauling assortments, uprooting and picking up stumps in a cutting area and transporting assortments to upper and intermediate timber warehouses.

The working hours of employees of the logging site are shiftable, and work is carried out around the clock: in the day and night shifts, the duration of the shift is 12 h, and in the middle of the fly-in period, workers can change shifts (from day to night and vice versa). The inter-shift rest period lasts 14 days.

All employees live on the territory of the Arkhangelsk region and reach the collection point (Karpogory, Yasny) by road or rail; from there, they are taken to the camp by bus along a forest road 100 km long. In the shift area, workers live in temporary cabins designed

for 4 people. These cabins are equipped with sleeping places and stoves and have a room for eating and relaxing. There is a canteen and a bathhouse on the territory of the camp.

The Arkhangelsk region of the Russian Federation was chosen for the study because forest harvesting is intensively developing in the north of the European part of Russia. A significant number of logging roads can be found here, as well as a few big rivers and large seaports of Arkhangelsk and Murmansk. The Arkhangelsk region is a powerful center of timber processing, which occupies a key place in the sectoral structure of the country. The region provides 8% of the total Russian volume of sawn timber, 29% of forest pellets, 26% of pulp and 17% of paper and cardboard [49]. All raw materials for woodworking are also produced on the territory of the Arkhangelsk region; in terms of forest reserves, the region ranks second in the North-Western Federal District. According to the Office of the Federal State Statistics Service for the Arkhangelsk Region and the Nenets Autonomous Okrug, the volume of timber harvested in 2020 amounted to 14.2 million m$^3$ (7% of all timber harvested in the country). This determines the relevance of this study for the Arkhangelsk region.

*2.3. Methods*

1.  The assessment of psychosocial factors was carried out using the general questionnaire of psychosocial factors at work (QPSNordic) [32,33,35], which includes an assessment of 123 psychosocial factors divided into 14 units. Responses to all items in both predictors were given on a 5-point Likert scale (very seldom or never; quite seldom; sometimes; quite often; very often or always). This diagnostic tool is valid and reliable [32,33,35]. Cronbach's alpha for questionnaire scales: personal background (0.76), job demands (0.65), role expectations (0.69), control at work (0.69), predictability at work (0.64), mastery of work (0.82), social interactions, social support (0.64), bullying and harassment (0.63), leadership (0.88), organizational culture (0.71), interaction between work and private life (0.68), commitment to the organization (0.87), group work (0.78) and work motives (0.75). The blocks of the questionnaire and sample questions are presented in Appendix A Table A1;

2.  Assessment of stress, working capacity and general functional state of employees.

One of the key areas of applied labor psychology is the development and practical implementation of psychoprophylactic programs and means to improve the efficiency of personnel and prevent the development of negative human functional states in various working conditions [50,51]. The human functional state (HFS) is defined as a relatively stable integrative pattern of activated internal resources (physiological, cognitive and emotional), which reveals the mechanisms of activity regulation in the current situation and determines the effectiveness of work in actual working conditions [51–53]. HFS study is based on work analysis data in order to identify sets of objective labor factors that lead to changes in HFS. By using this methodology, any type of HFS can be represented as a structural pattern of actualized physiological and mental functions involved in the process of performing work [52–54].

The characteristics of human functional states are indicators of the autonomic, endocrine and cardiovascular systems, measured in a certain period of time, as well as stress, working capacity and others. These characteristics reflect the number of physiological reserves available to a person and the degree of realization of potential opportunities for performing activities.

Traditionally, the technique of dynamic monitoring of human functional states involves three groups of methods [55–57]:

(1). Methods of instrumental psychophysiological diagnostics that are considered the most reliable since they register changes at the level of physiological and psychophysiological systems, in particular, analysis of heart rate variability and performance assessment based on sensorimotor reaction;

(2). Subjective–evaluative (questionnaire) methods that allow us to qualitatively study the state and mood of a person at the level of their subjective feelings and experiences;

(3). Psychological projective methods that allow assessing the human functional state through unconscious experiences.

Our previous study, devoted to the relationship between objective biochemical, instrumental psychophysiological, projective and subjective methods for assessing human functional states of shift oil and gas personnel in the Far North and in the Arctic, found that the maximum similar results in the assessments were obtained according to objective and projective parameters of states [57]; therefore, we chose the same methods in the present study.

2.1. Psychophysiological methods using the device for psychophysiological testing UPFT-1/30 "Psychophysiologist" (MTD Medikom, Taganrog, Russia):

"Variational cardiointervalometry" (VCM) is the assessment of the functional state and adaptive capabilities of the cardiovascular system by the variational cardiointervalometry method [57,58]. It helps to assess the general functional state of the body, in particular the state of the human autonomic nervous system, based on the analysis of electrocardiogram (ECG) parameters of the cardiac activity rhythm of the subjects [58]. During the examination, an ECG signal is recorded. The time between adjacent intervals between heartbeats (RR intervals) is measured. The minimum cycle for examination by the method is equal to the time required to register 128 cardiointervals. The integral indicator "the general functional state of body" is calculated on the basis of multiplicative convolution in accordance with the algorithm developed by "Nadezhnost" company (V.E. Kosachev, A.A. Talalaev) and evaluates the overall functional state of the human body according to heart activity parameters [58]. It allows identifying six types of the worker's states: critical, negative, maximum permissible, permissible, close to optimal and optimal, as well as the level of their functional capabilities (low, medium or high). The types of general functional states diagnosed by the device are presented in more detail in Table 2.

**Table 2.** Types of general functional state and its characteristics. Adapted with permission from Ref. [58] NPKF "Medicom MTD", 2017.

| № | Assessment of Functional State (VSR, rel. Units) | Functional State Type Name | Characteristic |
|---|---|---|---|
| 1 | >0.80 | optimal | The state of optimal tension of regulatory systems is necessary to maintain an active balance of the body with the external environment. The expenditure of the body's functional resources does not go beyond its capabilities. |
| 2 | >0.64 | close to optimal | A state of moderate tension in regulatory systems when the body needs additional functional reserves to adapt to environmental conditions. This condition occurs in the process of adaptation to work activity, during emotional stress or when exposed to unfavorable environmental factors. |
| 3 | >0.37 | permissible | The state of pronounced tension in regulatory systems, which is associated with the active mobilization of defensive mechanisms. |
| 4 | >0.1 | maximum permissible | The state of overvoltage of regulatory systems, which is characterized by the insufficiency of protective and adaptive mechanisms and their inability to provide an adequate response of the body to the effects of environmental factors. Excessive activation of regulatory systems is no longer supported by appropriate functional reserves. |
| 5 | >0.001 | (close to relaxed) | A state of depletion of regulatory systems, in which the activity of control mechanisms decreases (insufficiency of regulatory mechanisms). There is a destruction of the existing functional system, and performance decreases. Being in such a state, a person must suppress the need for rest by volitional effort. Therefore, his neuro-emotional stress increases, which manifests itself in a feeling of fatigue, irritability and negative emotions. |
| 6 | =0.001 | negative | It manifests itself in the form of a variety of functional disorders. Due to the significant need for rest, the regenerative functional system reaches a high level of power. In this case, a very intense volitional effort is required from a person in order to force himself to continue a certain activity. |

2.2. Psychological methods: The color preference test (M. Luscher, adapted by LN Sobchik) [59,60] was used, with the calculation of interpretation coefficients G.A. Aminev: working capacity and the presence of a stressful state [61]. On the basis of factor analysis, he singled out the following coefficients: heteronomy, concentricity, the balance of personality traits, the balance of the autonomic (vegetative) nervous system, working capacity and the presence of a stressful state. All these coefficients are calculated according to the corresponding formulas, reflecting a particular combination of colors [57]. In the present study, only the coefficient for assessing working capacity and stress was used. M. Luscher singled out a working group of colors (red, green and yellow). He argued that high performance corresponds to the preference of these colors by subjects. This feature is reflected in Aminev's coefficient of working capacity, which varies from 9.1 to 20.9. Indicators of stress, according to M. Luscher's theory, are finding the main active colors in the last positions of the choice and finding brown, black and gray cards in the first places of the row. Stress is measured from 0 to 41.8. Aminev's interpretation coefficients have been repeatedly used in empirical studies and are in good agreement with other parameters of human functional states [57,61].

As noted earlier, the characteristics of workers' psychophysiological functional states are some of the indicators of psychological safety and adaptation to professional activities [62,63]. In 2020, we studied the dynamics of loggers' functional states during a fourteen-day fly-in period using a set of objective and subjective assessment methods (monitoring was carried out daily in the morning and evening before and after the shift) [63]. We found out that the functional state level of workers in the evening in the first 4 days is acceptable, and on the fifth day in the morning, a negative state of workers was recorded. Since most employees have a high level of functional reserves, they have a sufficiently high potential and internal resources to overcome the negative impact of the professional environment, providing good adaptive opportunities. At the same time, against this high resource background, a decrease in the functional state to an acceptable level was observed. Its reduced level was observed in the morning on days 8 and 9, which is due to the change from day shift to night shift and vice versa. This requires employees to activate adaptive resources, which reduces their psychophysiological indicators. During the second week of the fly-in period, the employees' functional state-level slightly increases [63].

*2.4. Data Analysis*

Statistical processing of the results was carried out using descriptive statistics, frequency analysis and multiple stepwise regression analysis. Moreover, the statistical package IBM SPSS Statistics 23.00 was used.

Three multiple stepwise regression analyzes were used for dependent variables, such as the general functional state of the body, working capacity and stress, and independent variables, such as psychosocial factors at work. QPSNordic normalized scores were subjected to multi-regression analysis (Z-scores relative to individual arithmetic mean and standard deviation (interval from −3 to 3)). The quality of the regression model was assessed by the ANOVA criterion and the multiple determination coefficient. The normality of the residuals was controlled, and the collinearity of independent variables was excluded. The stepwise method was applied. Step selection criteria: probability F of inclusion—0.05, probability of exclusion—0.1.

The frequency analysis of employees' subjective assessments of psychosocial factors at work was carried out (point scale from 1 to 5), which demonstrated the nature of the distribution of assessments for individual items of the questionnaire. The analysis of the questions in the QPSNordic brought us to the conclusion that some of them have positive wording (for example, your immediate supervisor encourages you to speak out when you have different opinions) and therefore, the higher the score, the more often this manifests itself in work and, as a result, the impact on the employee is more favorable. Some of them are negative (for example, your work tasks are too difficult for you or you have been subjected to intimidation or harassment in the workplace during the last six months),

i.e., a high score on this parameter may mean the most adverse impact on the employee. Moreover, if there were factors that did not have a clear positive or negative wording (for example, work includes contacts with buyers or clients), as a rule, these are factors among the requirements for work and work organization. This can be determined only when correlated with the qualifications and motivation of the employee to perform this kind of work.

Next, we selected the factors of all three categories, which had the highest (5 and 4) or lowest (1 and 2) values in the majority of interviewed employees (more than 50%). Not only extreme estimates were chosen since some respondents had a tendency not to give them. We divided the selected factors into three groups:

(1). Labor factors (Table 10), which included neutral characteristics that describe the specifics of the conditions and organization of labor at the logging site;

(2). Positive psychosocial factors or an environmental resource (Table 11), which included the highest scores for factors with a positive wording (marked green in the tables) and the minimum scores for factors with a negative wording (marked red in the tables). Under the environmental resource, we understand the totality of external means that the subject possesses and uses to ensure effective activity and maintain an optimal human functional state in the process of adaptation;

(3). Negative psychosocial factors or risk factors, which included the highest scores for factors with negative wording and the lowest scores for factors with positive wording (Table 12).

The multiple regression analysis was used to determine the psychosocial factors in work that affect working capacity, stress and the general functional state of the body of forest harvesting personnel. At the same time, the frequency analysis was used to identify key positive and negative psychosocial factors.

## 3. Results

At the first stage of this study, the human functional state of loggers was assessed by objective psychophysiological instrumental (the VCM method using the "Psychophysiologist" device) and subjective methods (Table 3).

**Table 3.** Mean values and standard errors of the average general functional state of body, stress and working capacity of loggers.

| Name of Indicator | M ± SE | Level |
|---|---|---|
| General functional state of a body (VCM) | 0.60 ± 0.045 | acceptable |
| Stress (M. Luscher) | 7.91 ± 0.713 | low |
| Working capacity (M. Luscher) | 18.21 ± 0.151 | high |

We found out that loggers are characterized by an acceptable level of the general functional state of the body, high levels of working capacity and low stress levels.

Our data showed that 71.1% of employees have a favorable functional state and well-being, 28.9% unfavorable (according to the data of the VCM instrumental method). The average value of the employees' projective stress factor is 7.91, which indicates its low level, and the working capacity is 18.21, which corresponds to a high severity level. A general description of the loggers' functional states brings us to the conclusion that their psychological safety and adaptation to shiftwork in the Far North are quite high. Only 28.9% of employees are characterized by an increased risk of developing adverse states and well-being.

At the next stage of the study, three consecutive multiple stepwise regression analyzes were carried out, where the dependent variables included (1) the level of the general functional state of a body (according to the VCM method), (2) stress and (3) working capacity according to M. Luscher's method. The independent variable was the assessment of psychosocial factors at work according to the QPSNordic method.

The multiple correlation coefficients for the final models are statistically significant ($p < 0.001$, Table 4), so multiple regression models can be meaningfully interpreted.

**Table 4.** ANOVA for regression models.

| Model | Sum of Squares | Degrees of Freedom | Medium Square | F | Significance |
|---|---|---|---|---|---|
| Dependent variable is the general functional state of a body | 4.012 | 9 | 0.446 | 7.559 | <0.001 |
| Dependent variable is stress | 1784.205 | 5 | 356.841 | 7.431 | <0.001 |
| Dependent variable is the working capacity | 271.813 | 6 | 45.302 | 11.054 | <0.001 |

Multiple correlation coefficients correspond to the average strength of the relationship (from 0.57 to 0.70) and explain up to 43% of the dispersion of variables reflecting the general functional state of the body, stress and working capacity (Table 5).

**Table 5.** Summary of regression models.

| Model | R | R-Square | Corrected R-Squared | Standard Error of Estimation | Change Statistics | | | | |
|---|---|---|---|---|---|---|---|---|---|
| | | | | | Change R Squared | Change F | Degrees of Freedom 1 | Degrees of Freedom 2 | Significance Change F |
| Dependent variable is general functional state of a body | 0.700 | 0.489 | 0.425 | 0.24284 | 0.034 | 4.752 | 1 | 71 | 0.033 |
| Dependent variable is stress | 0.568 | 0.323 | 0.279 | 6.92959 | 0.044 | 5.070 | 1 | 78 | 0.027 |
| Dependent variable is working capacity | 0.680 | 0.463 | 0.421 | 2.02440 | 0.039 | 5.570 | 1 | 77 | 0.021 |

Checking the distributions of residuals for normality was carried out using the Kolmogorov–Smirnov test (Table 6).

**Table 6.** Kolmogorov–Smirnov criteria for the normal distribution of residuals.

| | Statistics | Degrees of Freedom | Significance |
|---|---|---|---|
| Dependent variable is the general functional state of a body | 0.055 | 81 | 0.200 |
| Dependent variable is stress | 0.097 | 84 | 0.051 |
| Dependent variable is the working capacity | 0.052 | 84 | 0.200 |

Note: Liljefors Significance Correction.

According to Table 7, the analysis of the regression equation demonstrates the following relationships: the simpler the work, the easier the tasks and the less overtime work, the more things employees manage to do, the more often there are interruptions in work that need to be eliminated and the more positive the general functional state of loggers is. All these parameters belong to the block of job demands. Due to the fact that the work of lumberjacks in most cases is monotonous, high employment and the need to eliminate minor disruptions can add variety to the activity, increase interest in the tasks performed and counteract monotony. At the same time, very complex tasks do not contribute to maintaining an optimal general functional state due to the need for additional mental efforts.

**Table 7.** Contribution of the assessments of psychosocial factors (coefficients of the regression equation) to the variability of loggers' general functional state.

| Model | Unstandardized Coefficients | | Standardized Coefficients | T | *p* Significance | 95.0% Confidence Interval for B | |
| --- | --- | --- | --- | --- | --- | --- | --- |
| | B | Standard Error | Beta | | | Lower Limit | Upper Limit |
| (Constant) | 0.644 | 0.053 | | 12.047 | <0.001 | 0.538 | 0.751 |
| JOB DEMANDS. Is your work challenging in a positive way? | −0.121 | 0.036 | −0.305 | −3.402 | 0.001 | −0.192 | −0.050 |
| JOB DEMANDS. Do you have too much to do? | 0.119 | 0.030 | 0.370 | 3.987 | <0.001 | 0.060 | 0.179 |
| LEADERSHIP. Does your immediate superior help you develop your skills? | 0.119 | 0.040 | 0.314 | 2.987 | 0.004 | 0.040 | 0.198 |
| MASTERY OF WORK. Can you yourself immediately asses whether you did your work well? | −0.087 | 0.045 | −0.170 | −1.921 | 0.059 | −0.177 | 0.003 |
| JOB DEMANDS. Are there interruptions that disturb your work? | 0.141 | 0.036 | 0.390 | 3.877 | <0.001 | 0.068 | 0.213 |
| ORGANIZATIONAL CULTURE. At your organization, are you rewarded (money, encouragement) for a job well done? | −0.129 | 0.036 | −0.351 | −3.625 | 0.001 | −0.200 | −0.058 |
| LEADERSHIP. Does your immediate superior encourage you to participate in important decisions? | 0.113 | 0.041 | 0.279 | 2.741 | 0.008 | 0.031 | 0.194 |
| JOB DEMANDS. Do you have to work overtime? | −0.086 | 0.035 | −0.230 | −2.425 | 0.018 | −0.157 | −0.015 |
| GROUP WORK. Is your group or team work flexible? | 0.046 | 0.021 | 0.195 | 2.180 | 0.033 | 0.004 | 0.088 |

The help of the immediate supervisor also contributes to a more positive general functional state of employees, helps them develop skills and participates in decision making. This can also be achieved by the flexibility of working in a team, as well as difficulties in self-assessment of the work performed and small rewards for a job well done. These correlations point to the importance of greater involvement from the immediate supervisor in the development of employees and their inclusion in decision making, which is also confirmed by the difficulties employees have in self-assessment of their work (they require external evaluation by management). At the same time, the correlation of remuneration for a job well-done with the general functional state may indicate the specific character of the material motivation system: high wages for the amount of work performed, while not much attention is paid to encouraging employees to work better.

According to Table 8, the regression equation for the dependent variable of working capacity includes the following relationships: the more work requires physical endurance and the higher the ability to influence decisions that are important for work, the more communications in the department, the less rigid organizational culture and also, the less the demands of a spouse or family interfere with work. This creates more opportunities to obtain support and help in work from colleagues, which increases loggers' work efficiency. Therefore, it should be noted that both labor factors (high physical endurance and the ability to influence important decisions, the absence of a rigid organizational culture) and factors of social support and understanding from family and colleagues have an impact on working capacity. In other words, the complexity of the activation process itself, corresponding to the abilities and qualifications of employees, and a comfortable social and organizational environment contribute to maintaining the optimal working capacity of personnel. Mutual understanding and optimal balance between work and private life are also important factors in working capacity, especially when working on a rotational basis when the husband is absent from home and carries out his professional activities in a forest plot for 14 days.

According to Table 9, the regression equation for the "stress" dependent variable identified a number of stressors: harassment and bullying in the workplace, a more rigid and rule-based corporate culture, lack of support for family members with difficulties in work, interruptions that interfere with work and lack of encouragement by one's direct superior to participate in decision making. Based on this, it can be argued that the factors of organizational culture and social support contribute more to the development of stress at work than work requirements. Possible family problems and help to families in difficult situations are important and cause stress reactions due to the specific character of shift work organization.

To test the second hypothesis, we analyzed the response frequencies of employees regarding all psychosocial factors at work. As can be seen from Table 10, the work of loggers with shift work organization is characterized by monotony, a fixed schedule and staying with a permanent group of colleagues without contact with clients, which requires physical endurance, maximum attention and great accuracy of movements.

**Table 8.** Contribution of the assessments of psychosocial factors (coefficients of the regression equation) to the variability of loggers' working capacity.

| Model | Unstandardized Coefficients | | Standardized Coefficients | T | $p$ Significance | 95.0% Confidence Interval for B | |
|---|---|---|---|---|---|---|---|
| | B | Standard Error | Beta | | | Lower Limit | Upper Limit |
| (Constant) | 15.914 | 0.428 | | 37.146 | <0.001 | 15.061 | 16.767 |
| INTERACTION BETWEEN WORK AND PRIVATE LIFE. Do the demands of your family or spouse/partner interfere with your work-related activities? | −1.019 | 0.319 | −0.286 | −3.190 | 0.002 | −1.655 | −0.383 |
| JOB DEMANDS. Does your work require physical endurance? | 1.064 | 0.273 | 0.328 | 3.893 | <0.001 | 0.520 | 1.608 |
| ORGANIZATIONAL CULTURE. Rigid and rule-based | −1.033 | 0.260 | −0.352 | −3.975 | <0.001 | −1.551 | −0.516 |
| ORGANIZATIONAL CULTURE. Is there sufficient communication in your department? | 0.811 | 0.272 | 0.271 | 2.981 | 0.004 | 0.269 | 1.352 |
| CONTROL AT WORK. Can you influence decisions that are important for your work? | 0.663 | 0.260 | 0.214 | 2.545 | 0.013 | 0.144 | 1.181 |
| SOCIAL SUPPORT. If needed, can you get support and help with your work from your co-workers? | 0.851 | 0.360 | 0.209 | 2.360 | 0.021 | 0.133 | 1.569 |

**Table 9.** Contribution of the assessments of psychosocial factors (coefficients of the regression equation) to loggers' stress variability.

| Model | Unstandardized Coefficients | | Standardized Coefficients | T | *p* Significance | 95.0% Confidence Interval for B | |
|---|---|---|---|---|---|---|---|
| | B | Standard Error | Beta | | | Lower Limit | Upper Limit |
| (Constant) | 10.611 | 1.733 | | 6.122 | <0.001 | 7.160 | 14.061 |
| BULLYING AND HARASSMENT. Have you noticed anyone being subjected to harassment or bullying at your workplace during the last six months? | 7.004 | 2.279 | 0.295 | 3.073 | 0.003 | 2.467 | 11.540 |
| SOCIAL SUPPORT. Do you feel that your friends/family can be relied upon for support when things become tough at work? | −4.171 | 1.127 | −0.382 | −3.702 | <0.001 | −6.414 | −1.928 |
| JOB DEMANDS. Are there interruptions that disturb your work? | −3.484 | 1.012 | −0.375 | −3.444 | 0.001 | −5.498 | −1.470 |
| ORGANIZATIONAL CULTURE. Rigid and rule-based | 2.264 | 0.864 | 0.251 | 2.620 | 0.011 | 0.544 | 3.984 |
| LEADERSHIP. Does your immediate superior encourage you to participate in important decisions? | −2.285 | 1.015 | −0.220 | −2.252 | 0.027 | −4.306 | −0.265 |

**Table 10.** General labor factors of loggers with shift work organization in the Far North (% of the total sample).

| Name of the Block of Factors | Name of the Factor | Very Seldom or Never | Rather Seldom | Some-Times | Rather Often | Very Often or Always |
|---|---|---|---|---|---|---|
| JOB DEMANDS. | Does your work require physical endurance? | 13.2 | 9.9 | 25.3 | 30.8 | 20.9 |
| JOB DEMANDS. | Does your work require maximum attention? | 5.7 | 3.4 | 11.4 | 34.1 | 45.5 |
| JOB DEMANDS. | Does your work require great precision of movement? | 6.7 | 3.4 | 12.4 | 37.1 | 40.4 |
| JOB DEMANDS. | Is your work monotonous? | 15.6 | 12.2 | 18.9 | 27.8 | 25.6 |
| JOB DEMANDS. | Do you have to repeat the same work procedure at intervals of a few minutes? | 7.8 | 7.8 | 11.1 | 32.2 | 41.1 |
| JOB DEMANDS. | Does your job include contact with customers or clients? | 96.6 | 3.4 | 0.0 | 0.0 | 0.0 |
| CONTROL AT WORK. | Can you set your own working hours (flexitime)? | 63.3 | 10.0 | 17.8 | 6.7 | 2.2 |
| CONTROL AT WORK. | Can you decide when to be in contact with clients? | 62.7 | 9.3 | 24.0 | 2.7 | 1.3 |
| GROUP WORK. | Do you appreciate belonging to this group or team? | 33.3 | 0.0 | 1.1 | 1.1 | 64.4 |

Based on Table 11, we can conclude that the enterprise where the study was conducted, in general, had created favorable working conditions and organizational culture, as evidenced by an extensive list of positive psychosocial factors at work. Employees have the necessary qualifications to solve professional problems, possess a clear understanding of the goals of their activities, positively assess their professional skills and receive the necessary support from colleagues. The organizational culture is favorable; there is no rigidity, competition, bullying and harassment, either from management or colleagues. The prevailing motivation of the staff lies in safety, calm life and material prosperity.

**Table 11.** Positive psychosocial factors in the work of loggers with shift work organization in the Far North (% of the total sample).

| Name of the Block of Factors | Name of the Factor | Very Seldom or Never | Rather Seldom | Some-Times | Rather Often | Very Often or Always |
|---|---|---|---|---|---|---|
| JOB DEMANDS. | Are your work tasks too difficult for you? | 46.1 | 24.7 | 23.6 | 4.5 | 1.1 |
| JOB DEMANDS. | Are your skills and knowledge useful in your work? | 6.7 | 12.4 | 0.0 | 48.3 | 32.6 |
| JOB DEMANDS. | Do you consider your work meaningful? | 10.1 | 3.4 | 16.9 | 32.6 | 37.1 |
| JOB DEMANDS. | Is it possible to have social contact with co-workers while you are working? | 4.5 | 3.4 | 23.6 | 38.2 | 30.3 |
| JOB DEMANDS. | Have you been exposed to threats or violence at work during the last two years? | 85.4 | 5.6 | 3.4 | 3.4 | 2.2 |

**Table 11.** *Cont.*

| Name of the Block of Factors | Name of the Factor | Very Seldom or Never | Rather Seldom | Some-Times | Rather Often | Very Often or Always |
|---|---|---|---|---|---|---|
| ROLE EXPECTATIONS. | Do you know what your responsibilities are? | 4.5 | 2.2 | 6.7 | 16.9 | 69.7 |
| ROLE EXPECTATIONS. | Do you know exactly what is expected of you at work? | 2.2 | 1.1 | 7.8 | 28.9 | 60.0 |
| ROLE EXPECTATIONS. | Does your job involve tasks that are in conflict with your personal values? | 44.4 | 25.6 | 16.7 | 7.8 | 5.6 |
| PREDICTABILITY AT WORK. | Is it necessary to demonstrate your ability and competence to others in order to be assigned to attractive tasks or projects? | 50.0 | 21.6 | 0.0 | 22.7 | 5.7 |
| MASTERY OF WORK. | Are you content with the quality of the work you do? | 3.3 | 3.3 | 18.7 | 54.9 | 19.8 |
| MASTERY OF WORK. | Are you content with the amount of work that you get done? | 2.2 | 4.4 | 22.2 | 46.7 | 24.4 |
| MASTERY OF WORK. | Are you content with your ability to maintain a good relationship with your co-workers at work? | 3.3 | 1.1 | 15.4 | 48.4 | 31.9 |
| MASTERY OF WORK. | Do you get information about the quality of the work you do? | 3.3 | 2.2 | 24.2 | 41.8 | 28.6 |
| SOCIAL SUPPORT. | Have you noticed any disturbing conflicts between co-workers? | 50.0 | 18.9 | 27.8 | 2.2 | 1.1 |
| BULLYING AND HARASSMENT. | Have you noticed anyone being subjected to harassment or bullying at your workplace during the last six months? | 98.9 | 0.0 | 0.0 | 1.1 | 0.0 |
| BULLYING AND HARASSMENT. | Have you been subjected to bullying or harrasment at the workplace during the last six months? | 98.9 | 0.0 | 0.0 | 1.1 | 0.0 |
| LEADERSHIP. | Is the relationship between you and your immediate superior a source of stress for you? | 46.0 | 25.3 | 20.7 | 5.7 | 2.3 |
| ORGANIZATIONAL CULTURE. | Distrustful and suspicious | 50.6 | 23.0 | 21.8 | 3.4 | 1.1 |
| ORGANIZATIONAL CULTURE. | Have you noticed any inequalities in how men and women are treated at your workplace? | 64.4 | 14.9 | 16.1 | 3.4 | 1.1 |
| ORGANIZATIONAL CULTURE. | Have you noticed any inequalities in how older and younger employees are treated at your workplace? | 71.3 | 14.9 | 12.6 | 1.1 | 0.0 |
| INTERACTION BETWEEN WORK AND PRIVATE LIFE. | Do the demands of your work interfere with your home and family life? | 58.0 | 19.3 | 17.0 | 2.3 | 3.4 |
| INTERACTION BETWEEN WORK AND PRIVATE LIFE. | Do the demands of your family or spouse/partner interfere with your work-related activities? | 64.8 | 17.0 | 17.0 | 0.0 | 1.1 |
| WORK MOTIVES. | To have good pay and material benefits | 2.4 | 2.4 | 7.1 | 33.3 | 54.8 |
| WORK MOTIVES. | To have a peaceful and orderly job | 6.1 | 1.2 | 13.4 | 43.9 | 35.4 |

**Table 11.** *Cont.*

| Name of the Block of Factors | Name of the Factor | Very Seldom or Never | Rather Seldom | Some-Times | Rather Often | Very Often or Always |
|---|---|---|---|---|---|---|
| WORK MOTIVES. | That the work is secure and provides regular income | 2.3 | 3.5 | 18.6 | 37.2 | 38.4 |
| WORK MOTIVES. | To have a safe and healthy physical work environment | 5.8 | 3.5 | 18.6 | 34.9 | 37.2 |

According to Table 12, risk factors for loggers are more related to their direct superiors, who do not always contribute to the development of their subordinates' skills, do not involve them in making important decisions, and do not encourage discussion of existing disagreements. Moreover, employees lack feedback on what skills and abilities they need to improve for further professional advancement.

**Table 12.** Negative psychosocial factors in the work of loggers with shift work organization in the Far North (% of the total sample).

| Name of the Block of Factors | Name of the Factor | Very Seldom or Never | Rather Seldom | Some-Times | Rather Often | Very Often or Always |
|---|---|---|---|---|---|---|
| PREDICTABILITY AT WORK. | Do you know in advance who will be your superior a month from now? | 36.7 | 17.8 | 13.3 | 15.6 | 16.7 |
| PREDICTABILITY AT WORK. | Do you know what is required in order for you to get a job that you consider attractive in 2 years? | 43.2 | 18.2 | 17.0 | 12.5 | 9.1 |
| LEADERSHIP. | Does your immediate superior encourage you to speak up when you have different opinions? | 37.1 | 20.2 | 29.2 | 9.0 | 4.5 |
| LEADERSHIP. | Does your immediate head encourage you to speak up when you have different opinions? | 37.1 | 15.7 | 39.3 | 4.5 | 3.4 |
| LEADERSHIP. | Does your immediate superior helpyou develop your skills? | 34.8 | 23.6 | 25.8 | 10.1 | 5.6 |

Our analysis revealed some regular patterns:

The following factors favorably influence the overall general functional state of loggers, having to perform many things, small interruptions in work, the help of the immediate supervisor in developing skills and intensifying participation in making important decisions and flexibility of working in a team. Unfavorable factors include the complexity of work, working overtime, the inability to independently evaluate the quality of the work performed and lack of remuneration for a job well done;

Loggers' working capacity is favorably influenced by such psychosocial factors as the need for physical endurance, the ability to influence decisions that are important for work and the opportunity to receive help and support in work from colleagues. Unfavorable influence occurs due to tough and rule-based organizational culture and the demands of the family interfering with the performance of work;

Harassment and bullying in the workplace, as well as a rigid and rule-based organizational culture, cause loggers' stress, while the following factors counteract the development of stress: the support of friends and family during hard work, interruptions that interfere with work and encouragement given by the immediate supervisor to participate in decision making;

Common labor factors for loggers include monotony, fixed work schedules and working in a permanent team without contact with clients, which requires physical endurance, maximum attention and great accuracy of movements;

The positive psychosocial factors of loggers include the availability of necessary qualifications for solving professional problems, a clear understanding of the goals of their activities, a positive assessment of their professional skills, the possibility of obtaining necessary support from colleagues and a favorable and comfortable organizational culture.

Risk factors for loggers include having a line manager who is not always conducive to developing the skills of subordinates, has little involvement in important decisions and does not encourage discussion of existing disagreements, as well as insufficient feedback on what skills and abilities need to be improved for further career and professional advancement.

## 4. Discussion

Our hypothesis is that the general functional state of the body, stress and working capacity, measured by objective and projective methods, will be differentially interconnected with psychosocial factors at work. Our study confirmed this hypothesis. This may be due to the fact that the general functional state of the body is more sensitive to hygienic microclimatic working conditions and physical stress, which is why this indicator is more closely related to work requirements. Therefore, according to these relationships, we can draw a conclusion regarding the optimality of workers' efforts spent in performing professional tasks in specific working conditions, as well as identify key factors that are more conducive to the strain of workers' adaptive resources.

Working capacity and stress, measured using a projective technique, reflect vaguely realized mental reactions and emotional manifestations associated with work. Therefore, we observed their greater correlation with factors of social support, leadership and organizational culture. The comfort provided by these conditions allows employees to feel greater satisfaction with their work and, as a result, increases their working capacity. Discomfort, on the contrary, contributes to the emergence of stressful conditions.

After evaluating the general functional state of the body, working capacity and stress by various methods (objective and projective) for the subsequent assessment of their relationship with psychosocial factors at work, we identified key labor factors that negatively affect employees, depleting their resources. Working in harmful conditions without appropriate PPE and labor protection measures can lead to a deterioration in workers' health. We also determined the psychosocial factors that increase working capacity and reduce stress due to comfortable socio-psychological and organizational working conditions. The choice of a single method for diagnosing the functional states of workers can reveal only one of the aspects of the analysis described above. Based on a comprehensive assessment, we developed targeted practical recommendations for the management of the enterprise and labor protection departments to improve personnel's working capacity, efficiency and safety.

Separate relationships between psychosocial factors at work and the human functional states of loggers obtained in this study were also confirmed by other authors. Hanse and Winkel [64] found that the variety of daily tasks, job rotation and access to breaks were statistically significantly associated with job satisfaction for forest machine operators. Their study also revealed a statistically significant positive correlation between work control and work rotation with reductions in musculoskeletal symptoms and between work rotation and flexibility of work breaks with reductions in headaches and sleep problems.

Mylek and Schirmer [15] found that workers who felt better in control of their jobs had better work-life balance and were more satisfied with their income, life in general and overall health. Psychosocial factors, such as job security, a positive workplace culture (involving the opportunity to express oneself), a perceived social support level, higher work efficiency and a positive work-related social identity, were also found to positively influence logger life satisfaction. Our study also found positive correlations between social support and the working capacity and stress of workers at a logging enterprise.

The results obtained suggest that psychosocial factors at work can be managed not only to reduce employees' stress but also to improve their well-being, engagement and commitment, as also noted by Maslach and Leiter [65]. It should be pointed out that

involvement contributes to the reduction in risky and unsafe behavior, adverse events, accidents and injuries [66]. This may be due to the fact that engaged employees pay more attention to the quality of work, show more concern for the quality of work and their professional reputation, and, therefore, are less inclined to unsafe behavior.

The data we obtained regarding the positive impact of support from the manager, colleagues and family members is of particular importance for these employees due to the specific character of the shiftwork method. These employees work for 14 days in the forest plots without returning home every evening. They are separated from their families. This makes the social support factor more important. Other studies also showed the role of psychosocial factors outside of work, which include marital status and family relationships. The potential for family conflict was found to reduce the cognitive resources of employees in the workplace [67]. This fact was previously noted by Rössler et al. [68], who also found a relationship between the well-being and marital status of workers.

Thus, successful management of psychosocial risks can lead to increased productivity [24] and create the opportunity for loggers to return home feeling better [24,29].

There is growing evidence that psychosocial factors [69] can have both positive and negative effects in the workplace. The risk factors identified by us, connected, first of all, with the relationship with the immediate supervisor, were reflected in the recommendations we proposed for the management of the enterprise. These factors need to be adjusted to improve the performance and well-being of all employees of the enterprise.

The list of positive factors that we consider as an environmental resource for employees, provided through an effective personnel management system of a logging enterprise, is quite extensive. Correlating the number of positive factors to risk factors at this enterprise brings us to the conclusion that employees are highly safe from a psychological perspective.

Our results are consistent with official statistics on the number of victims of accidents at work in the company where the study was conducted. There were 18 accidents in 2021, of which 0 were fatal, and one employee was in a state of alcohol intoxication (Table A2 Appendix A).

The results of a 14-day dynamic observation of the workers' general functional state of the body, working capacity and stress at this enterprise, which we conducted before this study [63], also confirm the conclusions made. According to objective, projective and subjective indicators of workers' functional states, their consistently favorable level is observed with multidirectional peaks during the shift change and a slight decrease at the end of the shift period. Operator performance is slightly higher in the second half of the shift period. Shift change is moderately stressful and is associated with changes in the body and psychology of workers, which is clearly manifested when measuring all the characteristics of their functional states. During the period of shift change, the risks associated with labor efficiency and safety increase, which undoubtedly requires the attention of the management of enterprises. The very fact of the change significantly increases stress and tension. It can be recommended to consider the option of employees working one shift (day or night) during the entire rotational period. At the same time, special attention should be paid to the fact that some employees have difficulty adapting to work at night.

The identified positive factors can be correlated with hygiene factors in the two-factor motivational theory of F. Herzberger [47,48], which are external in nature and relate to the context of work and not to its content. According to the author, such factors support health but do not always improve it. In order to improve it, motivational factors are needed that help employees achieve better results. Therefore, the control of these factors at an enterprise is an important task for the evaluation of preventive programs to maintain the required level of performance, health and well-being of workers.

The limitations of this study are the sample size and the lack of comparison with loggers in other regions in other natural and climatic conditions. Moreover, the range of methods for assessing functional states should be expanded. The present research has

promising new directions; for example, it would be interesting to apply this study to samples from other industries: oil and gas, diamond mining, construction, etc.

*Practical Organizational and Public Health Recommendations for Enterprise Management on Optimizing Staff Working Capacity and Performance*

Based on the results of the study, we offer the following recommendations and directions for optimizing the activities of shift personnel at logging enterprises in the Far North:

- By paying attention to the risk factors associated with the leader, it is recommended to take measures to improve the communication and managerial skills of masters with regard to the effective setting of tasks, giving feedback and motivating subordinates. Evaluation of workers' achievements directly by the manager is important for logging companies, which contributes to the fulfillment of professional duties at a high level. It is recommended to encourage the initiative of employees and discuss with them important decisions, so they can feel like an indispensable part of the team and be more committed to their organization;
- By analyzing risk factors with regard to work requirements, it should be noted that regulated breaks are one of the effective means of optimizing labor activity since the work of loggers is characterized by increased attention and greater accuracy of movements with increased responsibility for the performed work. Forest machine operators are forced to work in a monotonous environment. In this connection, it is important not only to provide the opportunity to independently choose the time for breaks but also to explain to employees the appropriateness of the breaks frequency and duration to restore their state and well-being, improve concentration and self-regulation during the shift;
- It is recommended to make changes in the system of motivation for a job well done. It is necessary to notice employees' successes; this will be an additional motivation to work more efficiently and better. It is required to provide explanations to employees on the existing system of their material and non-material motivation in order to increase their awareness and understanding of the principles of fairness and key working capacity indicators when assigning remuneration;
- For loggers, a good mobile phone or Internet connection is essential in the workplace with a shiftwork organization. While employees do not feel compelled to socialize, most workers have families and need to connect with them to feel safe and help them solve problems when needed.

The proposed measures to optimize labor organization and staff motivation increase the loyalty of employees to the organization, as well as their professional efficiency and labor safety because their job satisfaction grows, and they have a possibility of restoring internal reserves that are spent at work.

## 5. Conclusions

The general functional state of the body, stress and working capacity, measured by objective and projective methods, are differentially interconnected with psychosocial factors at work. The study shows that psychosocial factors have a different impact on the employee at different levels of his personality: from psychophysiological to socio-psychological. In this connection, it is important to ensure the comfort and safety of the employee's professional environment, which helps maintain his state of health and motivate him to work more efficiently. Psychosocial factors, interrelated in the general functional state of the body, contribute to maintaining the health status of the employee and are related to working capacity—to increase their motivation for more efficient work.

The following factors favorably influence the overall general functional state of loggers: a large number of tasks they have to perform, small interruptions in work, the help of the immediate supervisor in developing skills and intensifying participation in making important decisions and the flexibility of working in a team. Unfavorable factors include:

having to perform very hard work, working overtime, the inability to independently evaluate the quality of the work performed and the lack of remuneration for a job well done.

The working capacity of loggers is favorably influenced by such psychosocial factors as the need for physical endurance, the ability to influence decisions that are important for work and the opportunity to receive help and support in work from colleagues. We talk about unfavorable influence when the demands of the family interfere with work performance and rigid and rule-based organizational culture.

Harassment and bullying in the workplace, as well as a rigid and rule-based organizational culture, influence the development of loggers' stress. The following factors counteract the development of stress in employees: support of friends and family during difficult times at work, interruptions that interfere with work and encouragement to participate in important decisions given by the immediate supervisor.

The general functional state of forest harvesting workers is influenced by factors of labor content, intensity and organization. At the same time, the relationships with the immediate supervisor are important in order to increase the working capacity, efficiency of employees and their involvement in work. In order to maintain the working capacity of shift personnel who work away from home for a long time, it is important to ensure regular communication with family and friends.

In order to increase the working capacity and reduce the stress of forest harvesting personnel, measures are necessary to develop the communication and managerial skills of middle managers, to provide good Internet and telephone communications in forest plots to connect workers with families and optimize work and leisure regimes.

**Author Contributions:** Conceptualization, Y.K. and N.S. (Natalia Simonova); Data curation, N.S. (Nina Shadrina); Formal analysis, N.S. (Natalia Simonova); Funding acquisition, Y.K.; Methodology, Y.K.; Supervision, Y.K. and N.S. (Natalia Simonova); Validation, Y.K. and N.S. (Natalia Simonova); Visualization, Y.K. and N.S. (Nina Shadrina); Writing—original draft, Y.K. and N.S. (Nina Shadrina); Writing—review & editing, N.S. (Natalia Simonova). All authors have read and agreed to the published version of the manuscript.

**Funding:** Project number FSRU-2020-006 as part of the state task for fundamental research "Assessment of psychological risks in the professional activities of extreme specialists", 2020–2022.

**Institutional Review Board Statement:** The study was conducted in accordance with the Declaration of Helsinki, and approved by the Ethics Committee of Higher School of Psychology, Pedagogy and Physical Education, Northern (Arctic) Federal University, named after M.V. Lomonosov (protocol No. 2, 2021).

**Informed Consent Statement:** Informed consent was obtained from all subjects involved in the study.

**Data Availability Statement:** Certificate of registration of the database 2021621449, 5 July 2021. Application No. 2021621307 dated 24 June 2021. Dynamic study of the functional states of workers of a logging enterprise in the Far North during a rotational race.

**Conflicts of Interest:** The authors declare no conflict of interest. The funders had no role in the design of the study; in the collection, analyses or interpretation of data; in the writing of the manuscript, or in the decision to publish the results.

## Appendix A

**Table A1.** Blocks of the QPSNordic questionnaire.

| Block Name | Number of Questions |
| --- | --- |
| **1. Personal background** | 11 |
| **2. Job demands** | 26 |
| **3. Role expectations** | 7 |
| **4. Control at work** | 9 |
| **5. Predictability at work** | 12 |
| **6. Mastery of work** | 6 |
| **7. Social interactions**<br>**7.1. Social support** | 9 |
| **7. Social interactions**<br>**7.2. Bullying and harassment** | 3 |
| **8. Leadership** | 8 |
| **9. Organizational culture** | 13 |
| **10. Interaction between work and private life** | 2 |
| **11. Work centrality** | 2 |
| **12. Commitment to the organization** | 3 |
| **13. Group work** | 5 |
| **14. Work motives** | 7 |

Table A2. The number of victims of industrial accidents in the forest harvesting industry in the Russian Federation and the Arkhangelsk region.

| | All Enterprises of the Russian Federation 2021, Thousand People | Forest Harvesting in Russia, People | Forest Harvesting Arkhangelsk Region, People | Company Where This Study Was Conducted | | Company 2 (in the Same Region of the Country) | | Company 3 (in the Same Region of the Country) | |
|---|---|---|---|---|---|---|---|---|---|
| | 2021 | 2021 | 2021 | 2021 | 1 Half-Year 2022 | 2021 | 1 Half-Year 2022 | 2021 | 1 Half-Year 2022 |
| Average number of employees | | 39052 | 4910 | 2170 | 2460 | 414 | 460 | 418 | 423 |
| The number of victims of accidents at work, | | | | | | | | | |
| Total | 21.6 | 104 | 24 | 18 | 3 | 3 | 0 | 4 | 0 |
| men | 15.1 | 98 | 24 | 18 | 3 | 3 | 0 | 4 | 0 |
| women | 6.5 | 6 | 0 | 0 | 0 | 0 | 0 | 0 | 0 |
| through fault of employees and employer of organization | | 53 | 19 | 18 | 3 | 3 | 0 | 4 | 0 |
| workers under influence of alcohol or drugs | | 0 | 0 | 1 | 0 | 0 | 0 | 0 | 0 |
| of which are fatal | | | | | | | | | |
| Total | 1.21 | 5 | 0 | 0 | 0 | 0 | 0 | 0 | 0 |
| men | 1.12 | 5 | 0 | 0 | 0 | 0 | 0 | 0 | 0 |
| women | 0.09 | 0 | | 0 | 0 | 0 | 0 | 0 | 0 |
| through fault of employees and employer of organization | | 2 | 0 | 0 | 0 | 0 | 0 | 0 | 0 |
| women | 0.010 | | | | | | | | |
| Suspension from work due to alcohol and drug intoxication | | | | | | | | | |

Resource: Federal State Statistics Service of the Russian Federation, https://rosstat.gov.ru/working_conditions (accessed on 26 August 2022).

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
