# Peer review of "The Psychosocial Risk Factors Evaluation and Management of Shift Personnel at Forest Harvesting"

_forests, doi:10.3390/f13091447_

Round 1

Reviewer 1 Report

Please see the marked-up pdf. 

Author Response

Dear reviewer,

Thank you very much for your time and a positive attitude towards research!

All comments have been corrected.

All significant changes in the text of the article are highlighted in yellow.

  1. What do you mean through illustrating projective methods?

The M. Luscher color test was understood. The wording has been changed to: subjective.

  1. Please try to cited some quantitative results and mention some example for these three groups.

Would be nice if you would add practical conclusions for doing this study. I would suggest summarizing the abstract and attempt to address the above mentioned concerns.

The abstract has been modified to reflect the advice of all reviewers.

  1. The introduction is lengthy and I would highly recommend to shorten this part. Please directly pinpoint to the subject and avoid addressing widely through the manuscrip.

The research objectives are not clear for me.

Before the objective, you need to make conclusion on literature and previous research. By doing this the reviewer and a reader can find out what have been the drivers to motivate authors to embark this study.

Please elaborate and address these concerns. 

Shortened the introduction, added a clearer transition to the goal.

Thus, the analysis of scientific research revealed a high danger of working in the forest harvesting due to the specific conditions and organization of labor, the impact of the employee's functional state and well-being on labor safety (injuries, errors in work, etc.). The conducted literature analysis showed insufficient research on a comprehensive assessment of the psychosocial factors diversity that affect workers at forest har-vesting.

The above analysis indicates that studies often examine the negative states and experiences of employees, which include stress, which increase the risk of injuries and accidents in the workplace of forest harvesting workers. At the same time, it is important to assess the impact of the work on personal and on the development of positive psychological states, which include working capacity. The optimal working capacity allows employees to perform their tasks efficiently and on time. In this connection, it is important to determine not only the factors influencing the development of stress and overwork, for their subsequent control and reducing the impact on staff, but also the factors that contribute to improving working capacity and performance.

The assessment of psychosocial risk factors in conjunction with the diagnosis of the workers' state and well-being during the shift period will allow to determine the key organizational and personal conditions that affect the work safety of loggers and develop targeted recommendations for improving working conditions and increasing the efficiency of their professional activities.

  1. Summarize the hypotheses and tried to combined with the introduction.

Don't make a separate section.

Hypotheses have been adjusted, summarized and moved to the Procedures section at the request of one of the reviewers.

  1. Please form these criteria in the form of a hierarchical system to be able to see what criteria or factor placed at each level fo the hierarchical system.

In this section, we have listed the blocks of this questionnaire and given a link to the source of publication of its model and validation, which details how it was developed and which methodology was chosen. In order not to overload the article with additional information, we did not make more detailed descriptions of the questionnaire.

Please move this to the appendix section - Table moved to Appendix 1

  1. The result section must be summarized and written in a very clear way. Honestly that was not clear for me to judge about.

Added summary to the end of the Results section

  1. Please explain this last sentences and what are the possible reasons you would think of?

Added:

This may be due to the fact that engaged employees pay more attention to the quality of work, show more concern for their work and themselves as an employee, and thus are less inclined to engage in unsafe behavior.

  1. Shorten discussion of results

Shortened as much as possible, considering the recommendations of other reviewers.

  1. Reduce the conclusion section as well, and tell the reader which factor or set of factors were much important in this study. How your research is going to contribute into science?and how would be the practical outcomes of this research?

Reduced and redesigned

Best regards and with gratitude, the authors

Reviewer 2 Report

The article describes a relevant topic and proves that appropriate methods have been used. The statistical evaluation is well done and transparent. Until the end of the chapter Results, I was very content. But then the same arguments occur several times, the discussion is a bit weak.

Finally, the chapter Conclusion repeats what we already know. The reader might wish to learn more how to improve the situation of the workers and their performance, too. This asks from the authors to free themselves from their study and to contextualize it in a broader range. It would be fine if they could improve this chapter.

On line 296 et sec. the authors give interesting insights to patterns that depend on the time during the 14-days-stay and on the switch between the day and the night shift. Unfortunately, these patterns don’t occur later on in the paper. But maybe this can be a point for the conclusions, too?

Here are some minor corrections:

Line 361 the hyphens are not correct

Line 444 it must be 13 instead of 3

Table 5, and 7 “capacity” with c

Table 8 please translate the Russian question to English

Author Response

Dear reviewer,

Thank you very much for your time and a positive attitude towards research!

All comments have been corrected.

All significant changes in the text of the article are highlighted in yellow.

  1. The article describes a relevant topic and proves that appropriate methods have been used. The statistical evaluation is well done and transparent. Until the end of the chapter Results, I was very content. But then the same arguments occur several times, the discussion is a bit weak.

 Added and clarified discussions, added some generalizations. Due to the fact that one of the reviewers asked to reduce the discussion section by 3 times, we could not add more data without increasing the section.

  1. Finally, the chapter Conclusion repeats what we already know. The reader might wish to learn more how to improve the situation of the workers and their performance, too. This asks from the authors to free themselves from their study and to contextualize it in a broader range. It would be fine if they could improve this chapter.

Reduced and redesigned

  1. On line 296 et sec. the authors give interesting insights to patterns that depend on the time during the 14-days-stay and on the switch between the day and the night shift. Unfortunately, these patterns don’t occur later on in the paper. But maybe this can be a point for the conclusions, too?

Added:

The results of a 14-day dynamic observation of the workers' general functional state of the body, working capacity and stress at this enterprise, which we conducted on the eve of this study [63], also confirm the conclusions made. According to objective, projective and subjective indicators of workers' functional states, their consistently favorable level is observed with multidirectional peaks during the shift change and a slight decrease at the end of the shift period, operator performance is slightly higher in the second half of the shift period. It is shown that the shift change is moderately stressful and is associated with changes in the body and psyche of workers, which is clearly manifested when measuring all the characteristics of the functional states of loggers. It should be noted that during the period of shift change, the risks associated with the efficiency and safety of labor increase, which undoubtedly requires consideration by the management of enterprises. The very fact of the change significantly in-creases stress and tension. It can be recommended to consider the option of employees working in one shift (day or night) during the entire rotational period. At the same time, pay attention to the fact that some employees have difficulty adapting to work at night.

 Here are some minor corrections:

Line 361 the hyphens are not correct - corrected

Line 444 it must be 13 instead of 3 - corrected

Table 5, and 7 “capacity” with c - corrected

Table 8 please translate the Russian question to English - corrected

Best regards and with gratitude, the authors

Reviewer 3 Report

The work presented by the authors touches on an important element of forest processing. The results obtained could be contrasted with other hazardous occupations with similar factors. I feel that there is a lack of statistical data such as the number of accidents, the risk of using stimulants, etc. while working. The most objectively created survey will not be hard data (answers to some of the questions are a decision of the moment and often result from not knowing the whole survey).  

The abstract should be rewritten to meet the requirements of the journal (maximum 200 words). 

Proposals need to be redrafted. At this point, it is not appropriate to repeat all the results and discuss them again. The section for this is in the results and discussion which the authors for some reason broke into two sections. 

From verse 160 onwards, there is an elaborate research hypothesis along with the assumptions made in the paper. The assumptions should be separated into a separate segment called Assumptions. And in my opinion they should be in the methodology. The hypothesis should be simplified to have the simplest form.   

Regarding Table 12. has the research been collated with data from the labour inspector, the level of average wages for the industry, data on accidents at work, the number of people not allowed to work due to intoxicant consumption? This is an important element due to the fact that few people denigrate their employer even in classified surveys.

In the body of the paper there is no reference to Table 13 in the text. I suspect that an error has crept in at line 444 and there should have been 13 instead of Table 3. 

The discussion of the results lacks more juxtaposition with the literature. The chapter itself should be part of Chapter 3. which would allow discussion of results and juxtaposition with the literature. This would have been beneficial to the reader, who would not have had to go through the whole paper.  

Author Response

Dear reviewer,

Thank you very much for your time and a positive attitude towards research!

All comments have been corrected.

All significant changes in the text of the article are highlighted in yellow.

  1. The work presented by the authors touches on an important element of forest processing. The results obtained could be contrasted with other hazardous occupations with similar factors. I feel that there is a lack of statistical data such as the number of accidents, the risk of using stimulants, etc. while working. The most objectively created survey will not be hard data (answers to some of the questions are a decision of the moment and often result from not knowing the whole survey).  

Added in the introduction

Automation of production at forest harvesting had a positive effect on reducing the risk of injury, due to a decrease in the share of physical labor, the comfort of mechanized equipment. The accident rate for manual logging is 4 times higher than for mechanized forest logging, both in Louisiana [25], Sweden [26], and Russia [27]. At the same time, studies indicate the influence of the season of the year on the injuries of forest loggers: the highest intensity of accidents is observed in the summer [16], in January, February and March [4; 6; 28].

  1. The abstract should be rewritten to meet the requirements of the journal (maximum 200 words). 

Abstract reduced to 200 words and corrected.

  1. Proposals need to be redrafted. At this point, it is not appropriate to repeat all the results and discuss them again. The section for this is in the results and discussion which the authors for some reason broke into two sections. 

Due to the fact that, according to the requirements of the journal, two separate sections are required, we divided them separately into results and their discussion. After your comment, we tried to rework so that there were no repetitions. We hope we understood your comment correctly. Thank you!

  1. From verse 160 onwards, there is an elaborate research hypothesis along with the assumptions made in the paper. The assumptions should be separated into a separate segment called Assumptions. And in my opinion they should be in the methodology. The hypothesis should be simplified to have the simplest form.   

Hypotheses simplified and moved to procedure section

  1. Regarding Table 12. has the research been collated with data from the labour inspector, the level of average wages for the industry, data on accidents at work, the number of people not allowed to work due to intoxicant consumption? This is an important element due to the fact that few people denigrate their employer even in classified surveys.

Yes, they compared. Added table 2 to Appendix 1 and text to it in the discussion of the results.

The results obtained are consistent with official statistics on the number of victims of accidents at work in the company where the study was conducted, amounted to 18 people in 2021, of which 0 were fatal, 1 employee was in a state of alcohol intoxication (Table 2 Appendix 1).

  1. In the body of the paper there is no reference to Table 13 in the text. I suspect that an error has crept in at line 444 and there should have been 13 instead of Table 3. 

Yes, thanks, corrected

  1. The discussion of the results lacks more juxtaposition with the literature. The chapter itself should be part of Chapter 3. which would allow discussion of results and juxtaposition with the literature. This would have been beneficial to the reader, who would not have had to go through the whole paper.  

The discussion has been added and adjusted as much as possible and also considering the comments of other reviewers.

Best regards and with gratitude, the authors

Round 2

Reviewer 3 Report

I thank the authors for responding to my comments. I think the hypothesis should be simpler(one sentence double folded). I get lost in the presented hypothesis on a quick browse as it is a collection of relationships. A veritable logic gate showing that a grandmother with a moustache can be a grandfather if she satisfies 4 other factors. I would suggest a simplification. A good hypothesis is as general as possible. 

Thank you again for the corrections made. And I endorse for publication with a suggestion to change the hypotheses to simpler/shorter ones. 

Author Response

Dear reviewer, thank you for your recommendations and valuable feedback. We further simplified the hypotheses:

Hypotheses:

  1. The general functional state of the body, measured using the objective psychophysiological method, will have a strong correlation with the with psychosocial factors of requirements for work.
  2. The stress and working capacity, measured using the subjective psychological method, with psychosocial factors of social support, leadership and organizational culture.
  3. The frequency analysis of assessments of psychosocial factors at work allows us to identify positive and negative factors that affect the majority of employees of the forest harvesting enterprise.

With gratitude and best wishes, the authors